# Frost Heaving Damage Mechanism of a Buried Natural Gas Pipeline in a River and Creek Region

**DOI:** 10.3390/ma15165795

**Published:** 2022-08-22

**Authors:** Wenxian Su, Shijia Huang

**Affiliations:** School of Energy and Power Engineering, University of Shanghai for Science and Technology, Shanghai 200093, China

**Keywords:** buried pipeline, frost swelling, river & creek region

## Abstract

When the buried pipeline passes through the permafrost zone, the phenomenon of frost swelling occurs in the permafrost zone, which causes a certain degree of bending and deformation of the pipeline. As a result, the pipeline’s structural safety is compromised, and the pipeline finally fails during operation, posing a serious hazard to the natural gas pipeline’s operation. Whereas the theoretical research on soil frost heave is relatively comprehensive, the applied research on engineering problems is not yet complete. Therefore, it is necessary to predict frost heaving through experiments and numerical simulation, and put forward reasonable control measures for existing or potential problems. For the problem of pipeline damage caused by frost swelling of soil in the natural gas high-pressure regulator station in a river and creek region, the Drucker–Prager elastic-ideal plastic model of soil was selected for finite element analysis, and a reasonable finite element model of pipe-soil was established in this paper. Through the temperature field analysis, it was found that the soil around the buried pipe is affected by the pipeline and is lower than its freezing temperature, which makes the soil freeze and swell. Furthermore, through the thermal–structural coupling analysis, it was found that the buried pipe is affected by the freezing and swelling of the soil and the structure is greatly likely to be damaged. In addition, by analyzing the temperature distribution and frost heave deformation of the soil around the pipeline, as well as the deformation and force of the pipeline at different pipe temperatures, this paper also determined the ideal temperature for preventing frost heave damage to natural gas at high-pressure regulator stations as −1 °C. Finally, based on the results of the abovementioned analysis, the heating method was determined to improve the frost damage phenomenon at the high-pressure regulator. The results of the anti-frost and swell study were used to conduct field trials at natural gas high-pressure regulator stations where frost and swell had occurred. By adding heating furnace to increase inlet temperature, frost heaving of gas transmission pipeline can be effectively prevented. The results of the research provide a reference for both existing and new natural gas pipelines, and also accumulate experience for winter maintenance design and construction of pipeline engineering in seasonally frozen soil areas.

## 1. Introduction

The river and creek region has a well-developed water system, dense network of rivers, and soil with high water content. It is located in a soft soil area. The soil is mostly muddy silty, which is clayey and heavy with scattered sandy and saline soils, leading to instability [1]. The buried pipeline was manufactured according to the standards of the American Petroleum Institute: API 5L X52. Under the hydraulic gradient and temperature gradient, a permafrost area will be formed, and the soil around the buried pipeline will freeze and swell as well as melt and sink under the low temperature, which could cause damage. This article is based on a natural gas high-pressure regulating station in Shanghai to study the phenomenon of severe detachment of above-ground pipelines from their supports and ground rupture. The reason for pipeline damage caused by soil frost heave is that air temperature, natural gas pressure differences, and seasonal variation of the pipeline itself may produce dynamic freeze–thaw zones in foundation soil. In combination with the operation of the gas pipeline, the critical values for gas pipeline freezing and swelling can be calculated by numerical analysis, and the gas pipeline can be kept in a state of heat preservation by adding a heating furnace, which can prevent the pipeline from freezing and swelling caused by a sudden drop in temperature. The working area of the regulator station is mainly divided into two parts: the pressure transformation zone and the valve zone. The center of the pipeline in the pressure transformation zone is 1.0 m from the reference ground, and the main pipeline in the valve zone is buried at a depth of 1.946 m (as shown in Figure 1). The high-pressure regulator station is connected to the high-pressure gas network upstream, which filters, measures, regulates (depressurizes), and odorizes the high-pressure gas before delivering it to the downstream secondary high-pressure network.

When natural gas is transported through a sub-transmission regulator, a pressure drop occurs due to throttling and expansion of the channel cross-section, resulting in a sharp drop in gas temperature in the pipeline [2]. This phenomenon is also known as the Joule–Thomson Effect, also known as the Throttling Effect. This is a common phenomenon in the operation of natural gas pipeline pressure regulation. According to certain studies [3], the temperature of the gas pipeline drops by approximately 4 °C to 5 °C for every 1 MPa reduction in pressure.

The large difference in pressure forces frost to form on the surface of the pipes even in the hot summer months as shown in Figure 2. However, it does not affect the normal operation of the high-pressure station, so it does not attract enough notice. With the arrival of winter, the air temperature rapidly decreases, leading to a serious deformation of the pipeline network in the regulator station, such as the high rise of the ground with buried pipelines, and cracks on the ground (shown in Figure 3). Meanwhile, the above-ground pipes are seriously raised, as shown in Figure 4, and some of the pipes were detached from their supports and suspended up to 30 mm.

According to the recorded data, the freezing phenomenon occurred in winter when the high-pressure regulator station was built in the early stages, but it was not visible. Additionally, it did not affect the normal operation of the high-pressure regulator station. However, as the demand for natural gas has grown in recent years, the gas pressure from the upstream high-pressure gas pipe network has progressively increased to 5 MPa, while the gas pressure to the secondary pipe network is only 1.6 MPa. The reduction in gas temperature after changing pressure is more severe when there is a substantial pressure differential. The measured annual temperature of gas pipelines is −5 °C, and the lowest reaches −9 °C. In addition, the high-pressure station is located in the river and creek region, where the soil moisture content is as high as 28.7%. As a result, soil frost heave occurs around the buried pipe at low temperatures. As the weather becomes warmer, some of the frozen soil will melt and flow to fill the bottom of the pipeline. More, continuous repetition might cause the depth of a buried pipe to gradually decrease, making the pipeline constantly rise. The above-ground pipes are also gradually lifted off their supports. Furthermore, with the low winter temperature, the bends between the above-ground and buried pipes are subjected to extreme external loads due to soil freezing and swelling, making the normal operation of the pipelines unfeasible.

Frost heave hazards in buried pipelines at offtake stations are mainly related to two factors. First, the soil temperature around the pipe is below the freezing point, which is the dynamic condition that induces the occurrence of pipe frost heave. Second, the soil around the pipe is susceptible to frost heave and there is sufficient moisture in the soil to enable pipe frost heaving to occur [4]. This can cause uneven freeze-swelling or thawing deformations in the foundation soil of buried gas pipelines relative to soils without freeze–thaw processes. These deformations can affect the mechanical condition of the pipeline and seriously harm the buried natural gas pipelines [5,6,7,8]. In order to ensure the safe operation of natural gas pipelines, it is required to investigate the causes of above-ground pipeline detachment from supports and soil uplift at high-pressure regulator stations based on an analysis of soil freezing and swelling mechanics, and to propose effective solutions.

Soil frost heave is a multi-disciplinary issue. With the advancement of frost heave theory, more and more research has been conducted to integrate frost heave with practical engineering [9,10]. Over the years, the temperature distribution and force characteristics of buried pipes in frosty areas have been studied by experiments and numerical simulations, and a series of measures are put forward [11,12,13,14,15]. However, these studies are only limited to the water–thermal coupling field in the permafrost zone, which has certain guiding significance for the prevention of frost heave. But for the buried pipelines in the permafrost zone, the frost heave force is the real threat to the safety of the pipelines. Therefore, the coupling of water, temperature, and mechanical force in the permafrost zone should be given more attention [16]. The actual situation in the permafrost zone is complex and it is difficult to simulate frost heave under multiple coupled fields. However, with the development and application of finite element software in recent years, analysis of buried pipelines in permafrost zones has become increasingly popular [17,18,19]. The theory of frost heave for soils is currently being developed. At present, theoretical research on frost heave in soils is relatively comprehensive, but applied research on engineering problems is not. Thus, experimental and numerical simulations are needed to predict frost heave and propose reasonable control measures for existing or potential problems.

An elastic–plastic finite element model of the pipe-soil system was developed using the ANSYS software in this paper. Using this model, the mechanical response of a multi-year natural gas pipeline under the effect of frost heave was analyzed. The calculated results can provide a feasible solution to the frost heave phenomenon in natural gas high-pressure regulator stations and ensure the safe operation of high-pressure stations, which has a certain significance and guiding effect for practical industrial applications.

## 2. Method of Calculation

Soil freezing occurs when water freezes into solid ice, so soil particles need a certain amount of water content and the soil temperature must be below freezing temperature. Natural freezing and swelling have four stages: supercooling, jump, constant, and decreasing. Unlike metallic materials, soils are elastic–plastic materials with hardening or softening strain [20,21]. Thus, the deformation of the soil is influenced by many factors such as the anisotropy of the soil, the principal stresses, stress levels, stress paths, and stress state. The stress–strain relationship of soils is the key factor in the study of the various mechanical properties of soil [22]. Under external forces, the deformation properties of soils are mainly characterized by non-linearity, rheology, shear expansion, and anisotropy [23]. In order to study the deformation and mechanical characteristics of soil and establish the relationship between the stress, strain and time of soil, a kind of mathematical equations are proposed based on a series of engineering experience and related indoor and outdoor tests, which are called the constitutive model of soil.

In this paper, the most commonly used Drucker–Prager elastic-ideal plasticity model was adopted for the study of soils and the finite element method was used to study soil frost heave.

The basic assumptions of this paper are as follows:(1)The thermal expansion and contraction of soil particles caused by temperature changes were ignored, the soil particles were assumed to be rigid bodies, and only the change in volume of the soil caused by the freezing and expansion of water in the soil was considered.(2)The nuances of the soil structure were ignored and the soil is a single, homogeneous, continuous, and isotropic material.(3)No change of soil volume was considered during plastic deformation and the stress tensor sphere is zero.(4)The cohesive force of soil is greater than zero, that is, the soil is cohesive.

### 2.1. Equilibrium Equations of Temperature Field for the Frozen Soil

In the 1980s, Nixon [24] showed that heat transfer induced by heat conduction was two to three orders of magnitude higher than that induced by convection in the process of soil frost heave. Therefore, for permafrost, the change in the temperature field is mainly formed by its own heat conduction [25]. Thus, in the process of soil freezing, the control equation of temperature can be expressed as
(1)$$C\cdot \frac{\partial T}{\partial t}=\frac{\partial }{\partial x}\left(\lambda \cdot \frac{\partial T}{\partial x}\right)+\frac{\partial }{\partial y}\left(\lambda \cdot \frac{\partial T}{\partial y}\right)+L\rho \cdot \frac{\partial \theta }{\partial t}$$
where *T* is soil temperature; *C* is the specific heat capacity of the soil; *λ* is the thermal conductivity of the soil; *L*, *ρ*, *θ* are ice water latent heat of phase change, the density of ice in the soil, and ice content of the soil, respectively, and *t* represents time.

It is pointed out that the phase change of permafrost soil only occurs in a certain temperature interval (*T*_*m*_ ± ∆) [26]. We use the subscript u to denote unfrozen soil and f to denote frozen soil, and have
(2)$$C=\{\begin{array}{c} C_{u}\\ \frac{1}{2}\left(C_{f}+C_{u}\right)\\ C_{f} \end{array}\;\begin{array}{c} T>T_{m}+∆T\\ T_{m}-∆T\leq T\leq T_{m}+∆T\\ T<T_{m}-∆T \end{array}$$
(3)$$\lambda =\{\begin{array}{c} \lambda_{u}\\ \frac{1}{2}\left(\lambda_{f}+\lambda_{u}\right)\\ \lambda_{f} \end{array}\;\begin{array}{c} T>T_{m}+∆T\\ T_{m}-∆T\leq T\leq T_{m}+∆T\\ T<T_{m}-∆T \end{array}$$

For the soil around the buried natural gas pipeline in the river and creek region, the temperature at the underground constant temperature layer is constant. Heat is transferred between the surface and air, as well as between the buried pipe and the pipeline. The boundary conditions are defined by the following.

The initial temperature condition is defined by
(4)$${T|}_{z=z_{0}}=T_{0}$$

The condition on fixed boundary is defined by
(5)$$\lambda_{f}\cdot \frac{\partial T_{E}}{\partial n_{1}}=-q$$
where *z*_0_ is the location of the constant temperature layer of soil; *T*_0_ is the temperature at the thermostatic layer of the soil; *T_E_* is the external ambient temperature; *q* is the heat flux density constant, and ∂*n*_1_ is the normal vector with a fixed boundary.

### 2.2. Element Equilibrium Equations of Stress Field for the Frozen Soil

Considering soil frost heave and elastic–plastic strain at the same time, the total strain of frozen soil can be expressed as:(6)$$\left\{\varepsilon \right\}=\left\{\varepsilon_{e}\right\}+\left\{\varepsilon_{p}\right\}+\left\{\varepsilon_{h}\right\}$$
where {ε}, $\left\{\varepsilon_{e}\right\}$, $\left\{\varepsilon_{p}\right\}$, and $\left\{\varepsilon_{h}\right\}$ are total strain, elastic strain under external force, plastic strain under external force, and volume strain of soil due to phase change, respectively, of which,
(7)$$\left\{\varepsilon_{h}\right\}={\left[\varepsilon^{\upsilon }\;\varepsilon^{\upsilon }\;\varepsilon^{\upsilon }\;0\;0\;0\right]}^{T}/3$$
where $\varepsilon^{\upsilon }$ is the volume expansion strain of soil in the process of frost heaving, and it can be known from the frost heaving rate of soil that this part of the strain is the frost heaving rate of soil, of which,
(8)$$\varepsilon^{\upsilon }=\eta$$
and
(9)$$\left\{\varepsilon_{h}\right\}=\frac{\eta }{3}\left\{\beta \right\}$$
where
(10)$$\left\{\beta \right\}={\left[1\;1\;1\;0\;0\;0\right]}^{T}$$

We use $\left[D_{e}\right]$, $\left[D_{p}\right]$, $\left[D_{ep}\right]$ and  [B] to represent the elastic matrix, plastic matrix, elastic–plastic matrix and geometric matrix according to the law of plastic flow. Additionally, in the strain space, the equation of stress increment can be expressed as follows:(11)$$\begin{array}{cc} d\left\{\sigma \right\} & =\left[D_{e}\right]\left[\left\{d\varepsilon \right\}-\left\{d\varepsilon_{p}\right\}-\left\{d\varepsilon_{h}\right\}\right]\\ & =\left[D_{e}\right]\left[\left\{d\varepsilon \right\}-\left\{d\varepsilon_{p}\right\}\right]-\left[D_{e}\right]\left\{d\varepsilon_{h}\right\}\\ & =\left[D_{e}\right]\left\{d\varepsilon \right\}-\left[D_{e}\right]\left\{d\varepsilon_{p}\right\}-\left[D_{e}\right]\cdot \frac{\eta }{3}\left\{\beta \right\}\\ & =\left[D_{e}\right]\left\{d\varepsilon \right\}-\epsilon \left[D_{e}\right]\left\{\frac{\partial f}{\partial \sigma }\right\}df-\frac{E}{1-2\mu }\left\{d\varepsilon_{h}\right\}\\ & =\left[D_{ep}\right]\left\{d\varepsilon \right\}-\frac{E}{1-2\mu }\left\{d\varepsilon_{h}\right\} \end{array}$$
where
(12)$$\left\{d\varepsilon \right\}=\left[B\right]{\left\{d\delta \right\}}^{e}$$
(13)$$\left[D_{ep}\right]=\left[D_{e}\right]-\left[D_{p}\right]$$
(14)$$\left[D_{p}\right]=\frac{1}{A}\left[D_{e}\right]\left\{\frac{\partial f}{\partial \sigma }\right\}{\left\{\left[D_{e}\right]\frac{\partial f}{\partial \sigma }\right\}}^{T}$$
(15)$$A={\left(\frac{\partial f}{\partial \sigma }\right)}^{T}D_{e}\frac{\partial f}{\partial \sigma }-{\left(\frac{\partial f}{\partial \sigma_{p}}\right)}^{T}D_{e}\frac{\partial f}{\partial \sigma }-B$$
(16)$$\left[D_{e}\right]=\left[\begin{array}{cccccc} K+\frac{4}{3}G & K-\frac{2}{3}G & K-\frac{2}{3}G & 0 & 0 & 0\\ K-\frac{2}{3}G & K+\frac{4}{3}G & K-\frac{2}{3}G & 0 & 0 & 0\\ K-\frac{2}{3}G & K-\frac{2}{3}G & K+\frac{4}{3}G & 0 & 0 & 0\\ 0 & 0 & 0 & G & 0 & 0\\ 0 & 0 & 0 & 0 & G & 0\\ 0 & 0 & 0 & 0 & 0 & G \end{array}\right]$$
where *E*, *µ*, *f*, *ϵ*, *σ_p_*, *K* and *G* are respectively the elasticity modulus, elasticity modulus, yield function, plastic growth multiplier, plastic stress, bulk modulus and shear elasticity. *A* is a variable related to the mechanical properties and hardening coefficient of the material and to the yield criterion. Then the incremental form of node force is defined by
(17)$$\begin{array}{cc} {\{∆F\}}^{e} & =\int {\left[B\right]}^{T}\left\{∆\sigma \right\}dV=\int {\left[B\right]}^{T}\left[D_{ep}\right]\left[B\right]{\left\{∆\delta \right\}}^{e}dV-\int {\left[B\right]}^{T}\frac{E}{1-2\mu }\left\{∆\varepsilon_{h}\right\}dV\\ & =\int {\left[B\right]}^{T}\left[D_{e}\right]\left[B\right]{\left\{∆\delta \right\}}^{e}dV-\int {\left[B\right]}^{T}\left[D_{p}\right]\left[B\right]{\left\{∆\delta \right\}}^{e}dV-\int {\left[B\right]}^{T}\frac{E}{1-2\mu }\left\{∆\varepsilon_{h}\right\}dV\\ & ={\left[K_{s}\right]}^{e}{\left\{∆\delta \right\}}^{e}-{\left\{∆F_{h}\right\}}^{e} \end{array}$$
where the element stiffness matrix can be expressed as
(18)$${\left[K_{s}\right]}^{e}=∭{\left[B\right]}^{T}\left[D\right]\left[B\right]dV_{e}$$

The incremental form of the equilibrium equation of frozen soil element is expressed as
(19)$${\left[K_{s}\right]}^{e}{\{∆\delta \}}^{e}={\left\{∆F_{s}\right\}}^{e}+{\left\{∆F_{h}\right\}}^{e}$$
where ${\left[K_{s}\right]}^{e}$ and ${\left\{∆\delta \right\}}^{e}$ are the local tangential stiffness matrix of soil and local nodal displacement with increment format, respectively. ${\left\{∆F\right\}}^{e}$ and ${\left\{∆F_{h}\right\}}^{e}$ are the increment format of local nodal force caused by external load and the incremental form of local node force caused by soil frost heave, respectively.

### 2.3. Element Equilibrium Equation of Stress Field for Pipeline

The element equilibrium equation of stress field for pipeline is similar to that of soil, but the pipeline will not freeze and its total strain can be defined as:(20)$$\left\{\varepsilon \right\}=\left\{\varepsilon_{e}\right\}+\left\{\varepsilon_{p}\right\}+\left\{\varepsilon_{T}\right\}$$
where $\left\{\varepsilon_{T}\right\}$ is the volume strain due to temperature change
(21)$$\left\{\varepsilon_{T}\right\}=a\Delta T\left\{\beta \right\}$$
where a is the linear expansion coefficient and ΔT is the temperature increment.

Therefore, the incremental form of the pipeline element equilibrium equation is expressed as:(22)$${\left[K_{p}\right]}^{e}{\left\{∆\delta \right\}}^{e}={\left\{∆F\right\}}^{e}+{\left\{∆F_{T}\right\}}^{e}$$
in which, ${\left[K_{p}\right]}^{e}$ is the tangential stiffness matrix of pipeline element, and ${\left\{∆F_{T}\right\}}^{e}$ is the incremental form of nodal force caused by temperature change of pipeline.

Because of the good thermal conductivity and small wall thickness of the pipe material, its temperature change in the direction of wall thickness is not obvious, so the thermal stress is small. Therefore, the main force of the pipeline comes from the external load, and the external load of the buried natural gas pipeline comes from the frost heave of the soil.

### 2.4. Interaction between Pipeline and Soil

For the buried natural gas pipeline, the underground pipeline and the soil directly contact and interact, but they are not in unity, and their interaction is realized by friction contact. As for the three-dimensional pipe–soil contact model, the pipe surface is a hard surface. Therefore, the three-dimensional CONTA174 with an 8-node surface and surface contact element was selected, and each node of this element has three degrees of freedom, which can translate along the X, Y, and Z directions in the node coordinate system. The soil is a flexible surface, so the corresponding TARGE170 unit is selected.

Based on the finite element method of the soil and pipeline, the contact element stiffness matrix of the soil and pipeline and the corresponding element node force vector are superimposed in a global coordinate system, where:(23)$$\left[K\right]\left\{∆\delta \right\}=\left\{∆F\right\}+\left\{∆F_{h}\right\}+\left\{∆F_{T}\right\}$$

In this formula, [K], {∆δ}, and {∆F} are the global stiffness matrix, the global displacement at the moment of ∆t, and the incremental form of local node force caused by external load, respectively.

## 3. Calculation Model

### 3.1. Model Validation

The gas upstream of the natural gas regulator is pressurized up to 1.6 MPa by three transformations of the above-ground pipeline in the pressure transformation zone and transported out of the high-pressure regulator by the buried pipeline in the valve zone. As a result, the freezing and swelling phenomenon mainly occur in the valve zone [13].

In this paper, the above-ground pipeline and buried pipeline after the last level of pressure regulation were selected for modeling. Huang et al. [27] found that the finite element calculation method can better consider the factors of pipe–soil interaction when studying the pipe–soil interaction of river-bank pipeline, and the calculation results are in good agreement with the actual situation. The model also refers to the model established by Wu et al. [28] in studying the stress and deformation of buried oil pipelines in permafrost regions. In order to validate the rationality and reliability of the simulation, the simulation strain values at the same position were extracted for comparison (see Section 4.2.2). By comparing the measured values with the simulated values, the feasibility of the simulation is illustrated.

The river and creek region is located in a saturated soft soil area. The soil there is powdery clay with 28.7% water content, characterized by high water content, high porosity, high compressibility, low strength, and high sensitivity. The basic physical parameters of the soil are obtained from site reconnaissance and are shown in Table 1. The pipeline is made of X52 steel with the following chemical composition (mass fraction, %): C 0.14, Si 0.29, P 0.003, S 0.005, Mn 0.6, Ni 0.02, Mo 0.01, Cr 0.03, Fe balance. The yield strength of X52 steel is 360 MPa, and the ultimate tensile strength is 460 MPa. Ferrite and pearlite are the main components of this structure and the grain diameters range from 7 to 15 µm [29,30]. The Drucker–Prager model was used here. The geometric dimension of the model is 31 × 25 × 10 m. The finite element models of the gas pipeline pressure regulator station in river and creek region are shown in Figure 5, where the x direction is parallel to the end face of the buried pipeline, the y direction is axial to the buried pipeline, and the z direction is vertical. The material of the pipelines is X52 grade steel. The radius of all the joint elbows is R = 1.5 D. The center of the pipeline in the pressure transformation zone is 1.0 m from the reference ground, and the main pipeline in the valve zone is buried at a depth of 1.946 m.

The temperature distribution of the soil is a decisive factor for soil frost heave. In this paper, coupled thermal–structural simulation was used to study the frost heave damage of buried natural gas pipelines. That is to say, the temperature field of the soil was first calculated and then the temperature field was applied to the structural field as an initial condition.

The pipe–soil model is part of the high-pressure regulator station. The mechanical parameters of pipelines were obtained by consulting relevant regulations as shown in Table 2 [20]. The heat transfer between the outer surface of the above-ground pipeline, the support, and the air were ignored. The temperature drop of the gas inside the natural gas pipeline within 200 m long was negligible [31]. Therefore, the internal surface temperature of the pipeline was uniformly set to −9 °C as measured by the high-pressure regulator station. In mid-latitudes, the depth of the thermostatic layer of soil is 8–10 m, and its temperature is approximately the annual average temperature of the area [32]. The bottom temperature of the soil was set to 12.3 °C. The air temperature was chosen as the lowest temperature in the air in winter, which is 0 °C, and the heat-exchange coefficient between the soil surface and the air measured was 17.32 W/(m^2^·°C).

### 3.2. Study of Mesh Independence

In the finite element simulation process, the mesh division is of great significance. Firstly, the relatively rough grids were divided for preliminary calculation, and the grids were gradually encrypted under the condition that the trend of the trial results is basically correct. The results of multiple trials were compared. When the variation range is within the allowable range, it can be considered that the mesh has no effect on the calculation value. The number and type of meshes directly affect the accuracy of the calculation results. For the established pipe–soil model, most of its structure is relatively regular and can be divided into hexahedral mesh; while for the soil with buried bends, it can only be divided into tetrahedral mesh due to the complexity of its structure. In order to reduce the influence of the number of grid nodes on the calculation results and ensure the reliability of the calculation, this paper analyzed the mesh independence of the model and selected three groups of data. The number of grid nodes and the test results are shown in Table 3.

According to the comparison of the above table results, the difference between the grid test results of different densities was within the allowable error range. Therefore, the grid division of the model meets the requirements of grid independence, and it is considered that the grid had no effect on the calculation results. Considering the accuracy and calculation time, the calculation model selected the mesh with 1,121,153 nodes.

## 4. Results and Discussion

### 4.1. Thermal Analysis of Frost Damage in Buried Natural Gas Pipelines

The temperature distribution of buried natural gas pipelines and the surrounding soil at high-pressure regulator stations is shown in Figure 6. It can be seen that the overall pipeline temperature is −9 °C, which is lower than the ambient temperature due to the good thermal conductivity of the pipeline and low pipeline temperature. Therefore, the frosting phenomenon on the surface of the pipeline is well explained. The soil has a constant temperature of 12.3 °C. The temperature of the soil at the bottom was constant, while the temperature of the soil away from the pipe decreased uniformly from the bottom to the top, but remained above 0 °C. However, the soil around the buried gas pipeline had a lower temperature of 0 °C, and the closer to the pipeline, the lower the temperature. In addition, the temperature of the soil was symmetrically distributed around the pipeline, but the temperature of the soil at the bottom of the pipeline was lower than the temperature at the top due to the constant higher temperature at the bottom of the soil and the lower air temperature, with natural convection heat exchange between the top and the air.

### 4.2. Structural Analysis of Frost Damage in Buried Natural Gas Pipelines

The results of the simulated temperature field were used as the initial conditions for the structural analysis to carry out a coupled thermal–structural analysis of the pipe–soil model.

The inner pressure of the pipeline was 1.6 MPa, the coefficient of friction between the pipe and the soil was 0.31, and the acceleration of gravity in the vertical direction was applied to the whole model.

Axial constraints were applied to the buried pipe truncation and the rest of the direction is free. Soil is regarded as a semi-infinite gross, that is, the soil is fixedly constrained at the bottom, free at the top, and only axially constrained around. Frictionless constraints were set between the above-ground pipe and support.

#### 4.2.1. Analysis of Frost Deformation of Pipe-Soil Structures

The simulation results of the frost heave are shown in Figure 7a, and the model deformation can be isometrically enlarged to Figure 7b by selecting an appropriate angle for easy observation. It can be clearly seen that the soil in the above part of the buried pipe was raised up to 250 mm due to frost heave. However, for the above-ground pipe, the mating pipe was far away from the permafrost zone and there was no obvious displacement under its own gravity. However, the ascending pipe section was severely detached from its support, and the simulated results are consistent with the field.

The total displacement of the pipeline is shown in Figure 8. The displacement of the pipeline in the x, y, and z directions were extracted respectively. As shown in Figure 8a, in the x-direction, the underground pipe moved slightly due to soil frost heave, with a maximum displacement of 36 mm, which is caused by the asymmetry of soil frost heave and is not harmful in the x direction. In Figure 8b, there was almost no axial displacement change at the manifold place and buried ascending pipe. However, the displacement in the −y direction was as high as 172 mm at the over-ground elbow. This is because there are pipes buried in the soil in the +y direction. Additionally, the soil of over-ground frost heave was significant, while there was no buried pipe in the −y direction. Therefore, the volume expansion in the +y direction of soil around the over-ground elbow is much larger than that in the direction of −y. Moreover, in the z direction, it can be seen from Figure 8c that the manifold fell slightly under the action of gravity to stabilize on the support, but the buried ascending pipe thick section was displaced by nearly 120 mm in the +z direction due to soil frost swelling. That is to say, it was lifted, while the thin pipe section and the above-ground pipe were not. This caused a 28 mm lift in the above-ground pipe, with smaller displacements closer to the manifold.

#### 4.2.2. Comparison of Simulation Results with Measured Values

The strains of the pipeline in the above-ground bending section of the high-pressure regulator station were measured under frost and swelling conditions shown in Figure 9. In this case, the axial strains of the pipeline were measured at points 1, 2, and 4, and its radial strain was measured at point 3.

Simulated strain values at the same locations were extracted to compare with the measured ones shown in Figure 10. It can be seen intuitively that the measured and simulated values of pipeline strain changed according to the same trend, and the simulated values were always larger than the measured values because the temperature of the simulated pipeline was chosen as the lowest value of the measured gas temperature. However, the actual gas temperature fluctuated slightly with the change of gas flow. Moreover, affected by human operation, the positions of actual observation points and simulated observation points cannot be guaranteed to be absolutely the same, and there was a slight gap between the two. The maximum relative difference between the measured and simulated values was about 13.6%, which is within the allowable error range, illustrating the feasibility of this simulation.

## 5. Analysis of Temperature and Deformation

### 5.1. Frost Heave Analysis of Soils at Different Pipe Temperatures

The records of the natural gas high-pressure regulator station show that the soil temperature is not too low under natural conditions even in the winter, and almost no frost heave occurs. For the soil around the buried gas pipeline, the temperature will be affected by the pipeline temperature, and frost heave will occur. Therefore, the thermal–structural coupled simulation of the pipe–soil structure was carried out at −6 °C, −3 °C, −2 °C, −1 °C, 0 °C, 1 °C, 2 °C, and 3 °C, respectively. It was also used to determine the optimum temperature of the soil frost heaving pipe in the high-pressure regulator station to ensure its safe operation.

#### 5.1.1. Temperature Distribution of the Surrounding Soil at Different Pipe Temperatures

The change in the pipe temperature causes a corresponding change in the soil temperature. Here, four sections were selected: the end of the buried ascending pipe (section A), the section where the buried ascending pipe joints the underground bend (section B), the section where the inlet ascending pipe joints the underground bend (section C), and the upper surface of the soil model (section D). The soil was extracted from three points (Figure 11) above and below the pipeline at sections A and B, and from three points around the pipeline at section C and D (Figure 12) respectively. For sections A and B, the six points were distributed from top to bottom as A1, A2, A3, A4, A5, and A6, B1, B2, B3, B4, B5, and B6, and for sections C and D, the six points were distributed from right to left as C1, C2, C3, C4, C5, and C6 and D1, D2, D3, D4, D5, and D6, for a total of 24 points.

For the soil around the ascending pipe section of the buried pipeline, the variation of soil temperature around the pipeline with the pipeline temperature at sections A and B is shown in Figure 13 and Figure 14. As the pipeline temperature rose, so did the soil temperature. In addition, for the soil below the pipeline (A4, A5 and A6 and B4, B5, and B6), the closer the point was to the pipe, the lower the temperature. The temperature difference between the three points decreased as the pipe temperature rose. For the soil above the pipe (A1, A2, and A3 and B1, B2, and B3), when the pipe temperature was below 0 °C, the temperature closer to the pipe was lower, and the difference in temperature at three points gradually decreased with the increase in pipe temperature, until the pipe temperature reached 0 °C and the temperature at three points was almost the same. When the temperature of the pipe rose above 0 °C, the temperature difference between the three points appeared again. The temperature nearer the pipe was higher. As the pipe temperature was always lower than the soil constant temperature layer, the difference in temperature between the pipe and the soil thermostatic layer was larger, causing the soil temperature below the pipe to rise rapidly. However, as pipe temperature rose, the temperature difference between the pipe and the soil thermostatic layer decreased.

After the soil temperature reaches the freezing temperature, there was always a temperature gradient for the soil beneath the pipe shown from Figure 13, Figure 14 and Figure 15. However, when the pipe temperature was above −1 °C, the soil did not freeze and the buried ascending pipe was not lifted. As for the soil above the pipe, when the pipe temperature reached −1 °C, the soil temperature was still below the freezing temperature. However, the differences in temperature between the two adjacent nodes were small, all within 0.5 °C. This means that the gradient in the soil temperature distribution is small. So, while soil will freeze at this time, the amount will be minimal. However, when the pipe temperature reaches 0 °C, the soil temperature is higher than the freezing temperature, so the soil will not freeze at this time.

In regard to the soil around the ascending section of the buried pipe, changes in soil temperature at section C and Section D with the pipe temperature are shown in Figure 16 and Figure 17. It can be seen that the temperature variations of soil with pipeline temperature in Figure 16 and Figure 17 were similar to those above the pipeline in Figure 13 and Figure 14. Soil temperature on the left side of the pipe was obviously higher than 0 °C in section C. In addition, for the soil on the right side of the ascending pipe, the soil temperature far away from the ascending pipe was about 0 °C under the influence of the buried pipe and the surface temperature. At section D, which is the surface of the land, the soil temperature away from the pipe was constant at about 0 °C.

In addition, when the temperature of the pipeline was below 2 °C, the temperature of the soil on the right side of the pipeline was significantly lower than that on the left. This is due to the buried pipeline under the soil on the right side of the straight pipe. The pipeline and the surface ground temperature affect the temperature of the soil away from the pipeline. However, there is no buried pipe under the soil to the left of the ascending pipe, so the temperature of soil far away from the ascending pipe is only affected by the constant temperature layer of the soil and the surface temperature. Therefore, with a temperature of less than 2 °C, the soil frost heave occurs more on the right than the left side, causing uneven force on both sides of the pipe, pushing it in the −y direction.

Hence, in order to prevent frost heaving of the soil around the inlet ascending section, it is sufficient to ensure that no frost heave of the soil on the right of the inlet ascending pipe occurs. In order to observe the influence of the pipe temperature on the soil temperature on the right-hand side of the inlet ascending pipe more clearly, the soil temperatures on the right side of the inlet ascending pipe can be compared at both sections. As shown in Figure 18, it can be seen that the change in soil temperature near the inlet ascending pipe is similar to that of soil temperature on the lower side of the buried ascending pipe. When the pipe temperature was −1 °C, although the soil temperature was below the freezing temperature, the temperature difference between the two adjacent nodes was small, all within 0.5 °C. At this time, soil frost heave occurred, but the amount of frost heave was small. Moreover, when the pipeline temperature reached 0 °C, the soil temperature was higher than the freezing temperature and no freezing and swelling occurred.

#### 5.1.2. Analysis of Freezing and Deformation of Soil at Different Pipe Temperatures

The soil around the buried ascending pipe and the soil around the inlet ascending pipe were selected in this paper without regard for gravity to study the changes in soil frost heave with the pipe temperature. Additionally, Figure 19 and Figure 20 illustrate the results of analyzing the change in soil vertical displacement due to changes in pipe temperature.

The soil displacement below the buried ascending reflects the height of the buried ascending pipe being lifted, while the soil displacement above it represents the observed ground soil uplift height. It can be seen that the height of buried ascending pipe and the height of ground uplift were gradually reduced with the increase of pipeline temperature. When the soil reached 0 °C, there was no frost heave. However, when the pipe temperature reached −1 °C, the change of soil displacement below the buried ascending pipe was about 0, that is, there was no obvious frost heave and the pipe is not lifted. At this time, the soil surface above the buried ascending pipe uplifted about 20 mm, which could be ignored when placed in a wide area of soil.

For the ascending pipe, when the pipe temperature was below 0 °C, the soil displacement on the right side was always greater than that on the left, indicating that the frost heave on the right side of the ascending pipe was greater than that on the left. When the pipeline temperature reached 0 °C, the soil displacement differential between the left and right sides of the pipeline was zero, and there was no frost heave. When the pipeline temperature dropped to −1 °C, the soil on both sides of the ascending pipe still had obvious frost heave, but the difference in vertical displacement was within 20 mm, indicating that the frost heave on both sides was similar. Therefore, the influence on the pipeline should be analyzed in terms of stress.

In conclusion, when the pipe temperature was 0 °C, no freezing and swelling occurred; when the pipe temperature was −1 °C, frost heave occurred in the soil, but the amount of frost heave was small.

### 5.2. Analysis of Frost Heaving Damage of Pipe-Soil Structure under Different Pipe Temperatures

In order to determine the optimum temperature for safe of pipelines, the frost heaving damage of pipe–soil structures at −1 °C and 0 °C were analyzed, respectively.

#### 5.2.1. Deformation Analysis of Pipes at Different Pipe Temperatures

The displacement variation along the pipeline at −1 °C is shown in Figure 21. In this case, the pipeline displacement in the x direction can be ignored. The displacement in the y direction is substantial, especially the displacement of the above-ground elbow connected to the ascending pipe to the ground which reached −12 mm. That is, the left and right forces of the pipe were uneven and moved in the y direction. In the z direction, the pipeline moved downward as a whole. Frost heave raised the earth below the buried ascending pipe by about 1 mm, while the soil around the buried ascending pipe sunk by about 3 mm due to frost heave.

The displacement variation in each direction when the pipeline temperature was set at 0 °C is shown in Figure 22. Under this condition, the displacement of the pipeline in both the x direction and the y direction was small and negligible. In the z direction, the pipeline moved downward as a whole, and the displacement in the z direction was close to that when under the condition of ignoring frost heave. That is, the displacement of the pipeline in the z direction is the result of gravity.

#### 5.2.2. Force Analysis of Pipes at Different Pipe Temperatures

When the pipeline temperature was at −1 °C, the pipeline had a certain displacement change under the action of soil frost heave. The stress of the pipeline was investigated at 1 °C and 0 °C.

When the pipeline temperature was −1 °C, the stress distribution of the pipe is shown in Figure 23. The overall stress distribution of the pipeline was small, but there was still a stress concentration area.

The pipe stress distribution at 0 °C is shown in Figure 24. Currently, there is no frost heave in the soil. The maximum overall stress in the pipe was 311MPa at the intersection of the manifold, which is similar to the stress distribution of the pipeline neglecting the frost heave.

By comparing the stress distribution of the pipeline at two kinds of temperatures, it can be seen that the stress of the above-ground pipeline and the underground pipeline was slightly larger than that at 0 °C when the pipeline temperature was −1 °C. In addition, the maximum stress points were located at the junction of the manifold, and the difference was not large.

When the pipeline temperatures were set at −1 °C and 0 °C, the primary membrane stress of the pipeline was about 17 MPa, and the primary membrane stress plus the secondary membrane stress was about 18 MPa, which is far less than the limit value. Thus, the high-pressure regulator station only needs to ensure that the temperature of the buried pipeline reaches −1 °C.

## 6. Improvement of Frost Heave in High Pressure Regulator Station by Heating

The temperature of the pipeline in the high-pressure regulator station is the key to soil frost heave. Moreover, the temperature of the pipeline is affected by the temperature of the natural gas in the pipeline. According to the frost heave damage analysis of the pipe soil structure under different pipe temperatures, the natural gas temperature after the last stage of pressure regulation is required to reach −1 °C.

### 6.1. Calculation of Temperature Drop of Gas in the Pipeline after Pressure Regulation

For pipeline natural gas, when the volume content of methane is greater than 85%, the Joule Thomson coefficient and the natural gas temperature after pressure regulation can be calculated according to the following formula [33]:(24)$$T_{2}=T_{1}-\overset{ˇ}{\mu_{J}}\left(P_{1}-P_{2}\right)$$
(25)$$\overset{ˇ}{\mu_{J}}=C_{P}\left(\frac{0.980\times 10^{6}}{T_{1}{}^{2}}-1.5\right)$$
where $T_{1}$ is the temperature of natural gas in the pipeline before throttling and $T_{2}$ is the temperature of natural gas in the pipeline after pressure regulation; $P_{1}$ is the pressure of natural gas in the pipeline before throttling and $P_{2}$ is the pressure of natural gas in the pipeline after throttling; $\overset{ˇ}{\mu_{J}}$ is the Joule Thomson coefficient; $C_{P}$ is the constant pressure mass-specific heat of natural gas before throttling.

### 6.2. Heat Load Calculation of Natural Gas Heating System and Selection of Heating Furnace

The commonly used heating system is to add a natural gas heating furnace to heat the natural gas before entering the pipeline.

When the flow is constant in the high-pressure regulator station, the heat load of the required heating furnace is calculated as follows:(26)$$q=Q_{m}C_{P}∆t$$
(27)$$Q_{m}=Q\rho$$
where q is theoretical heat exchange rate; ∆t is the temperature rise; $Q_{m}$ is natural gas mass flow; Q is volume flow of natural gas; ρ is Natural gas density.

In addition, since the actual throttling process is an irreversible process under unsteady conditions and there exists energy loss, the actual heat exchange rate should be multiplied by a coefficient of 1.2. Thus, the heat load of natural temperature rise is 1.2q [34].

According to the numerical simulation results, the natural gas was introduced into the main pipeline inlet of the station after being cut off and heated according to the numerical simulation results of the main pipeline inlet, and then the natural gas was introduced into the main pipeline inlet. In addition, a maintenance bypass valve group was set in the overall heating equipment.

A 60 × 10^4^ Nm^3^/h natural gas hot water heating furnace was set up in the heating furnace system of the high-pressure regulator station to meet the final heating load. In addition, natural gas was used as the fuel gas, and the gas pressure was lower than 0.4 MPa. The gas consumption was adjusted according to the daily transmission capacity of the high-pressure regulator station. The site diagram is shown in Figure 25.

At low temperatures in winter, when the temperature after pressure regulation in the station is lower than −1 °C, the natural gas heating furnace will be opened. The temperature controller maintains the water temperature in the furnace body at the set temperature and controls the combustion system as required to heat the water. The heating device was used for a period of time, and there was no deformation in the field, which achieved the expected effect of preventing frost heave.

## 7. Conclusions

The frost heave of a natural gas high-pressure regulator station was studied considering soil frost heave temperature field, the interaction between the soil and pipeline, thermal structure coupling, gas temperature decline, and system heat load. The critical temperature of the pipeline frost heave was obtained by numerical calculation, and the improvement of adding a heating furnace was put forward. The main results are as follows:(1)The temperature of the soil around the buried pipeline was far lower than that of the soil away from the pipeline at the same level. The closer it was to the pipeline, the lower the temperature was. Additionally, it was lower than the freezing temperature of the soil, resulting in a frost heave of the soil.(2)Because of soil frost heave, the primary local membrane stress plus secondary stress at the maximum stress point of the pipeline was very close to the allowable value. Considering the soil freezing and thawing cycle caused by seasonal change, the pipeline structure would be easily destroyed.(3)It was found that the pipeline could operate safely at both −1 °C and 0 °C. The high-pressure regulator station should ensure that the buried pipeline temperature reaches over −1 °C.(4)By adding a heating furnace and increasing the inlet temperature, the frost heave of the gas transmission pipeline could be effectively prevented.

## Figures and Tables

**Figure 1 materials-15-05795-f001:**
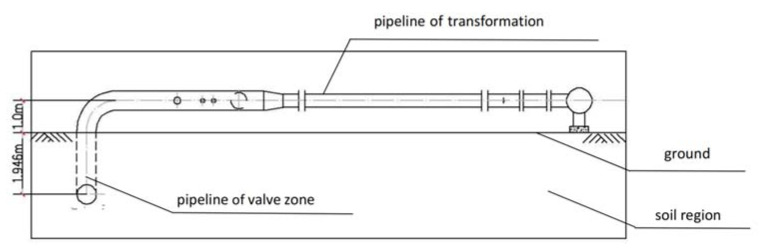
Diagram of the main pipeline in the valve zone.

**Figure 2 materials-15-05795-f002:**
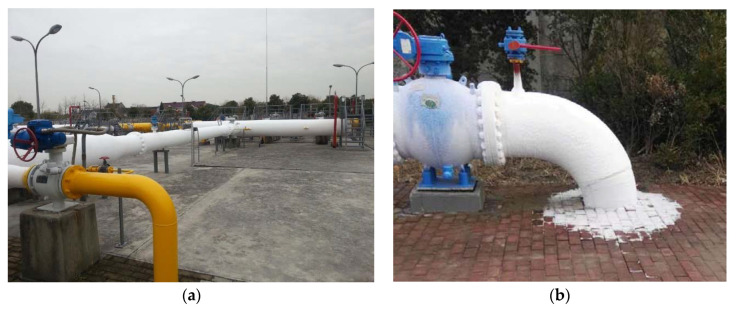
Frosting of pipes on the site: (**a**) frosting of the above-ground pipeline; (**b**) frosting of the pipeline into the ground.

**Figure 3 materials-15-05795-f003:**
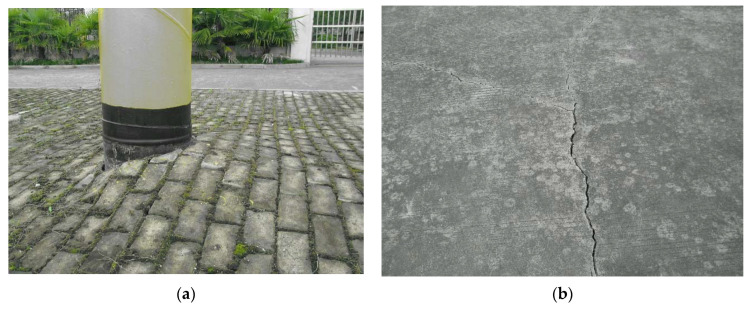
Ground conditions on the site: (**a**) the ground heaved at the pipeline into the ground l; (**b**) the ground heaved and cracked at the buried pipe.

**Figure 4 materials-15-05795-f004:**
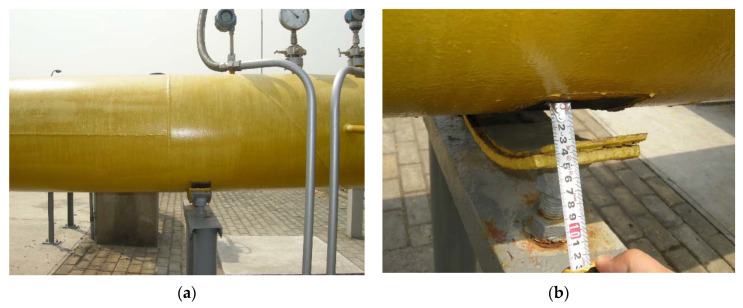
Diagram of pipe breakaway support on the site: (**a**) general view of the pipe disconnecting bracket; (**b**) enlarged view of the pipe disconnecting bracket.

**Figure 5 materials-15-05795-f005:**
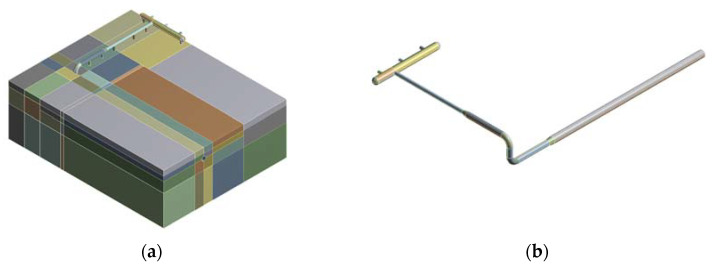
Element model diagram: (**a**) Overall model diagram; (**b**) piping model diagram.

**Figure 6 materials-15-05795-f006:**
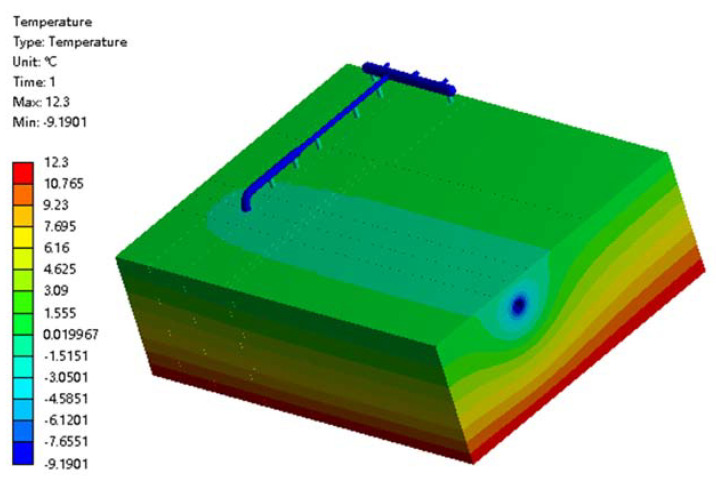
Cloud map of temperature field in the buried pipeline and the surrounding soil.

**Figure 7 materials-15-05795-f007:**
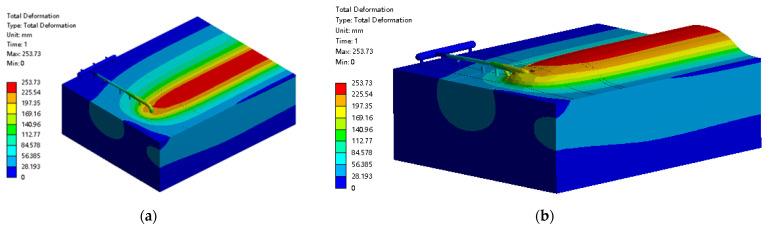
Overall frost heave damage diagram of buried pipeline and its surrounding soil: (**a**) cloud diagram of frost deformation of pipe–soil structures; (**b**) isometric cloud diagram of the frost deformation of the pipe–soil structure.

**Figure 8 materials-15-05795-f008:**
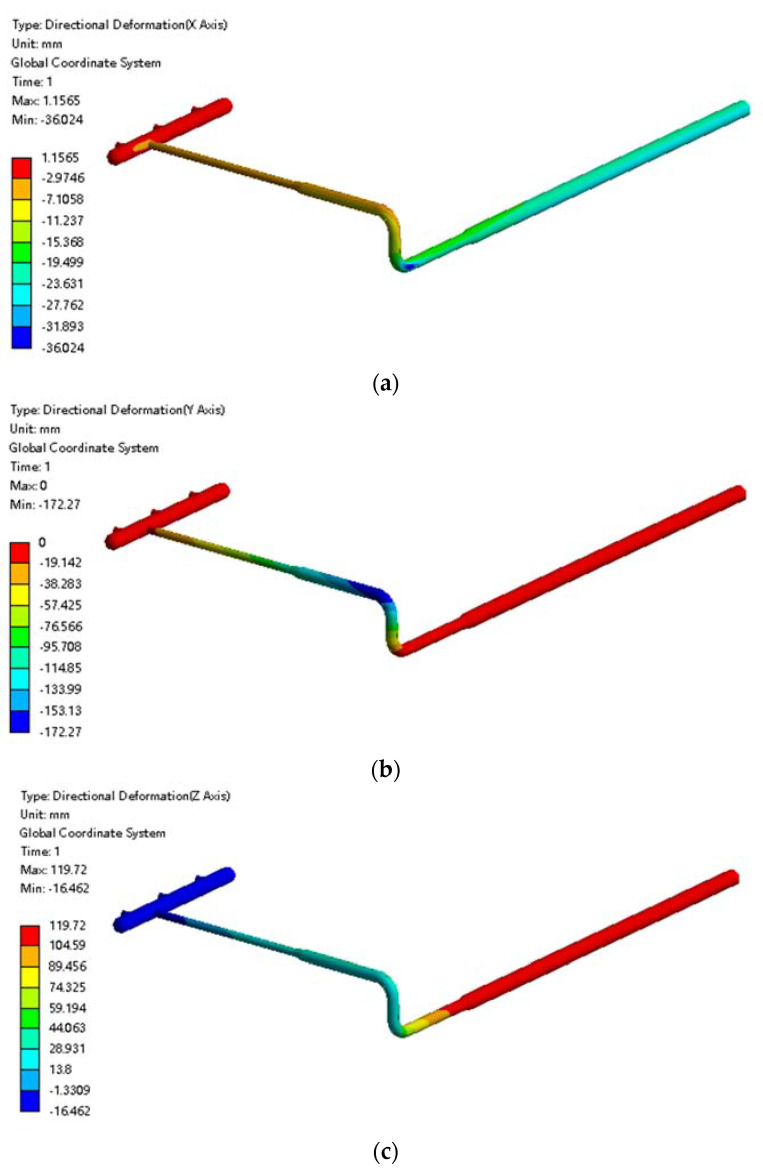
Cloud diagram of the pipeline displacement under frost heave: (**a**) Displacement cloud diagram of the pipeline in the x direction; (**b**) displacement cloud diagram of the pipeline in the y direction; (**c**) displacement cloud diagram of the pipeline in the z direction.

**Figure 9 materials-15-05795-f009:**
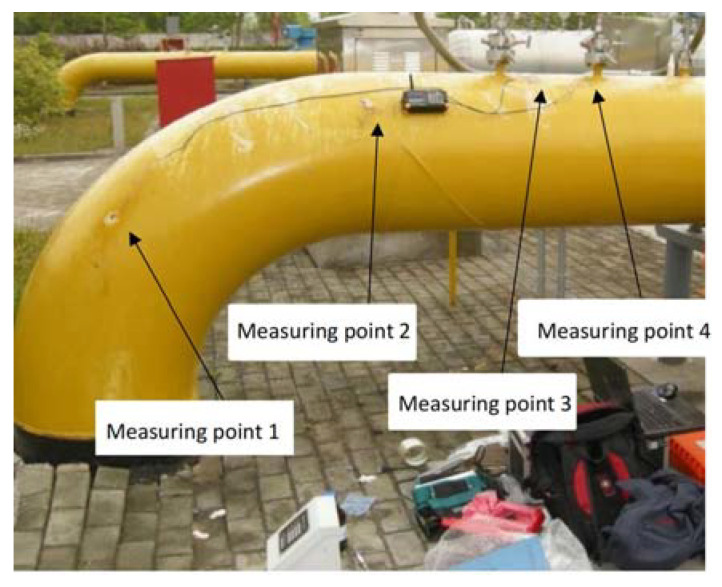
Distribution of field measuring points.

**Figure 10 materials-15-05795-f010:**
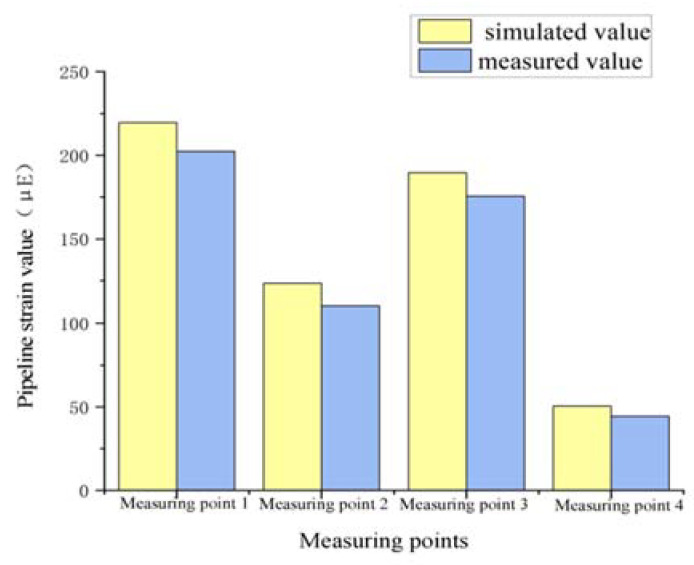
Comparison of pipe strain values.

**Figure 11 materials-15-05795-f011:**
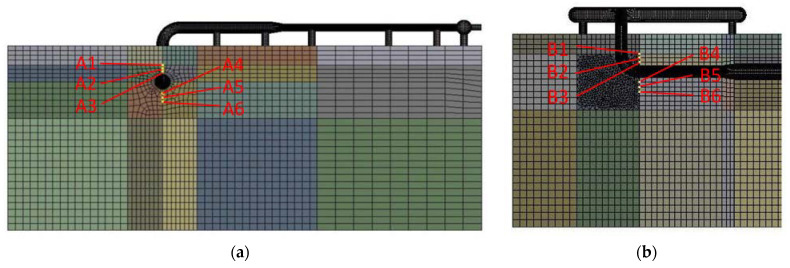
Plots of soil temperature around buried ascending pipe: (**a**) section A; (**b**) section B.

**Figure 12 materials-15-05795-f012:**
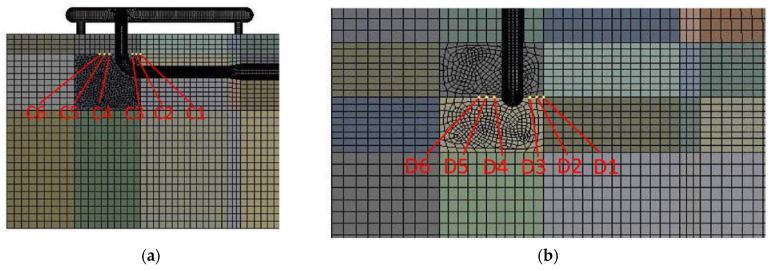
Plot of soil temperature around the ascending pipe into the ground: (**a**) section A; (**b**) section B.

**Figure 13 materials-15-05795-f013:**
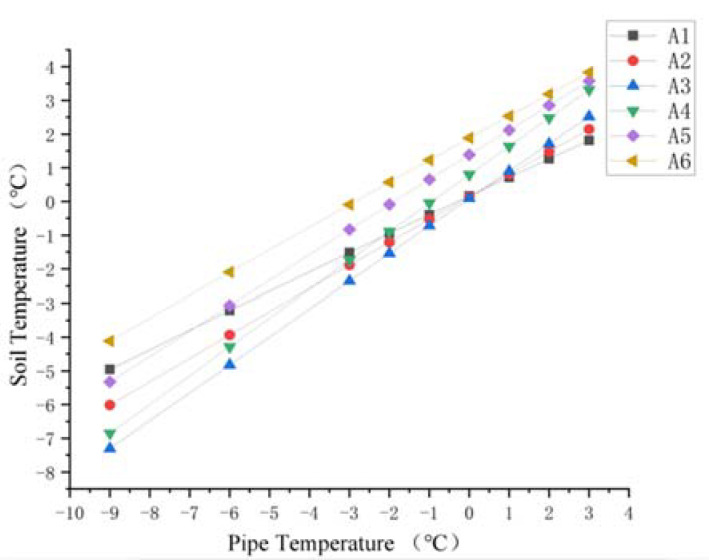
Diagram of soil temperature trend at point A1–A6.

**Figure 14 materials-15-05795-f014:**
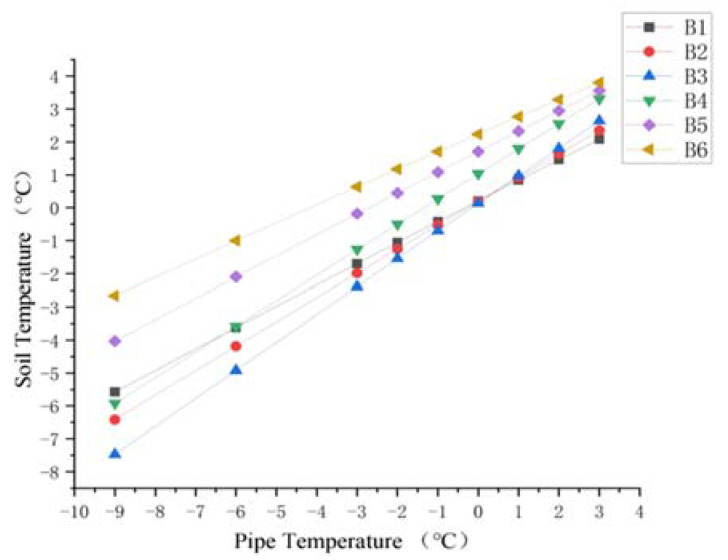
Diagram of soil temperature trend at point B1–B6.

**Figure 15 materials-15-05795-f015:**
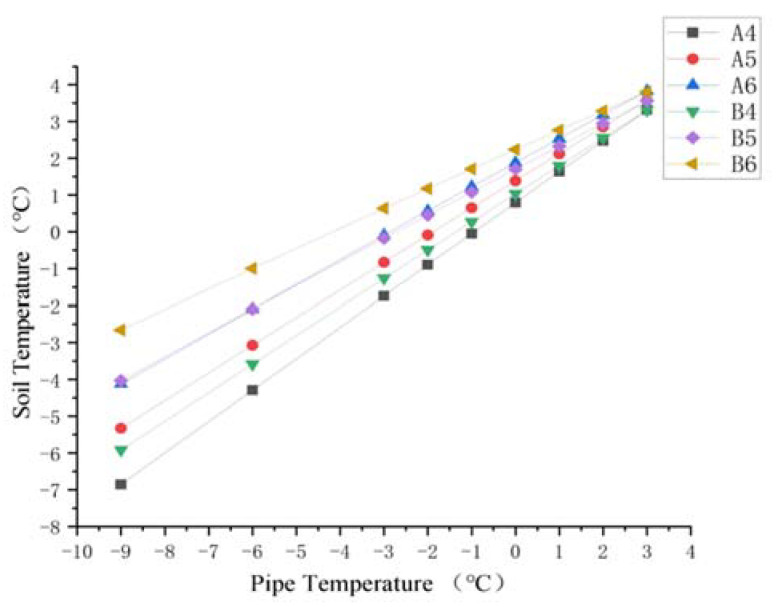
Diagram of soil temperature trend for pipes beneath cross-section A and B.

**Figure 16 materials-15-05795-f016:**
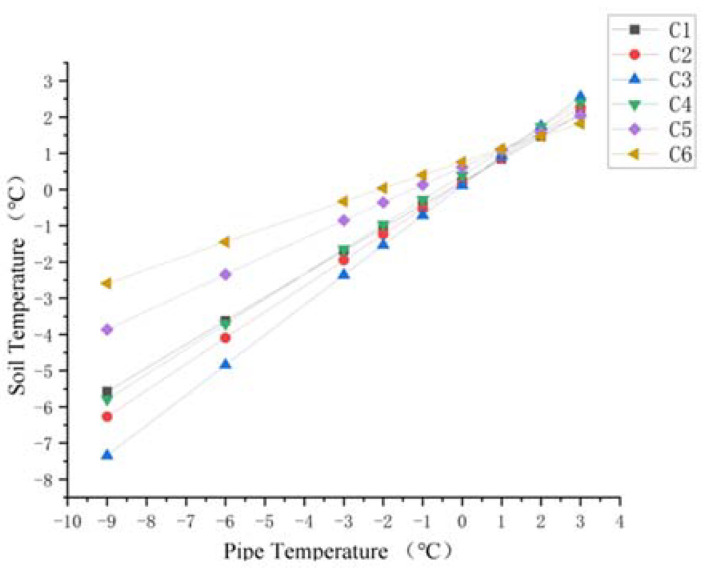
Diagram of soil temperature trend at point C1–C6.

**Figure 17 materials-15-05795-f017:**
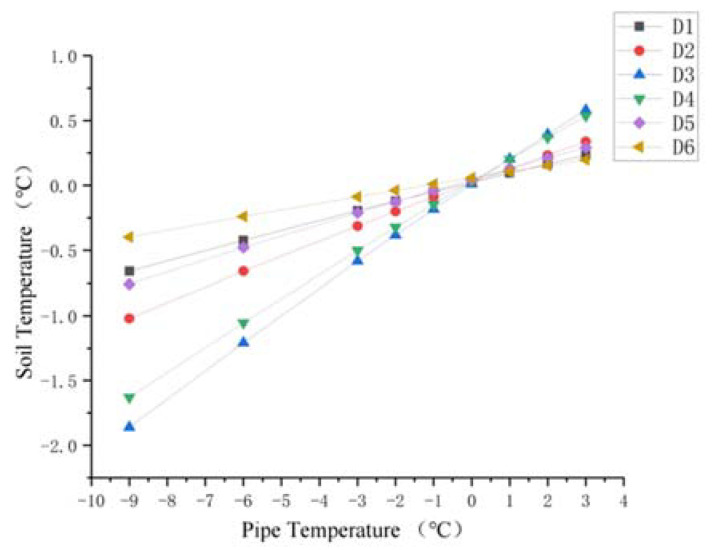
Diagram of soil temperature trend at point D1–D6.

**Figure 18 materials-15-05795-f018:**
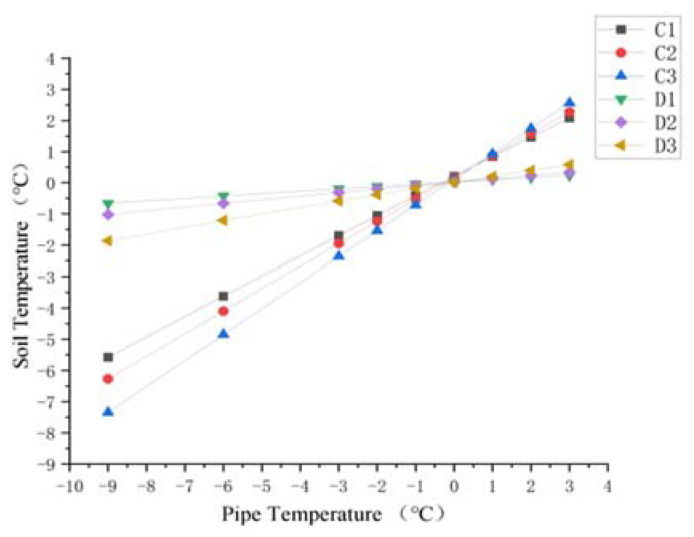
Diagram of soil temperature trend on the right side of the inlet ascending pipe.

**Figure 19 materials-15-05795-f019:**
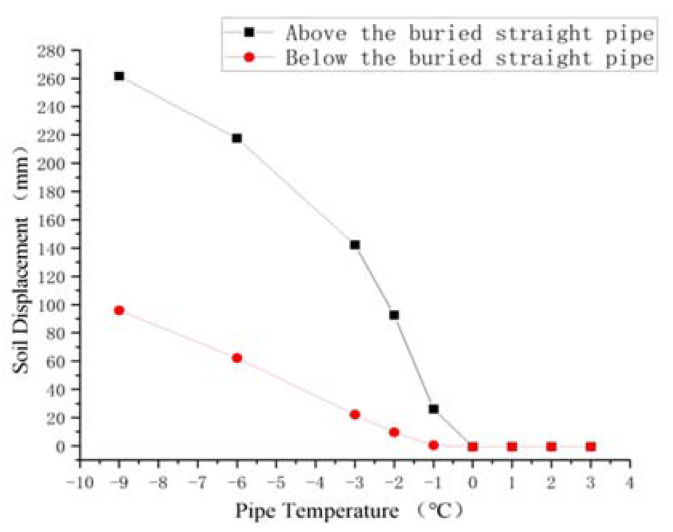
Frost heave displacement diagram of soil near the buried ascending pipe with pipe temperature variation.

**Figure 20 materials-15-05795-f020:**
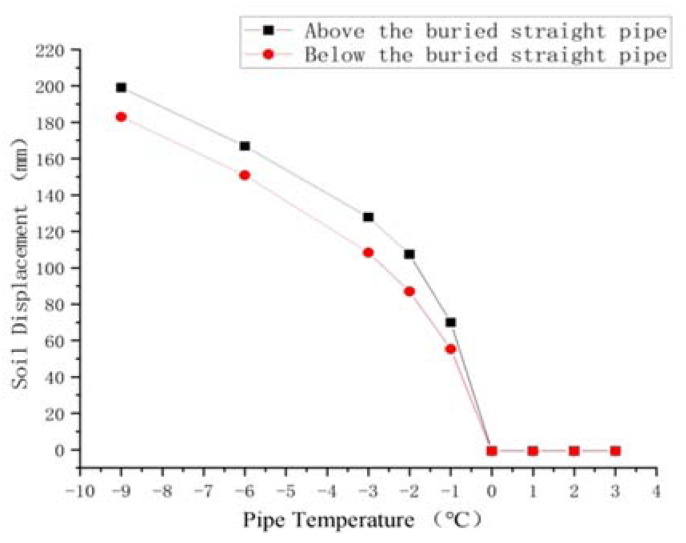
Frost heave displacement diagram of soil near the inlet ascending pipe with pipe temperature variation.

**Figure 21 materials-15-05795-f021:**
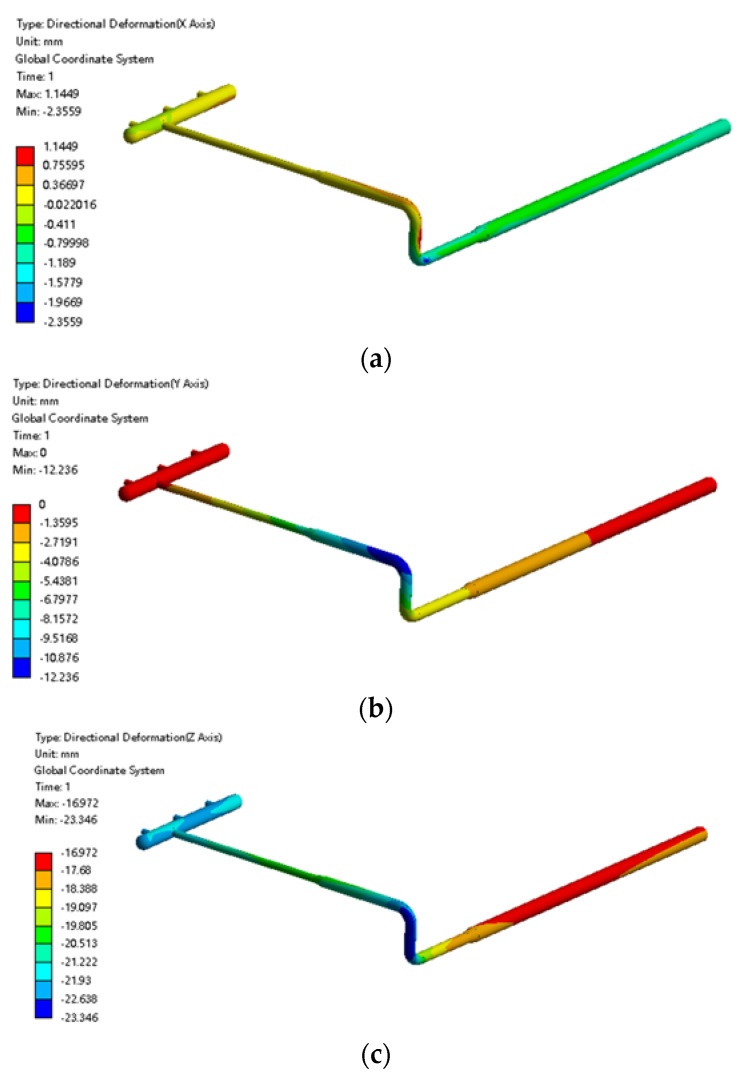
Cloud diagram of pipeline displacement at −1 °C: (**a**) cloud diagram of pipeline displacement at the x direction; (**b**) cloud diagram of pipeline displacement at the y direction; (**c**) cloud diagram of pipeline displacement at the z direction.

**Figure 22 materials-15-05795-f022:**
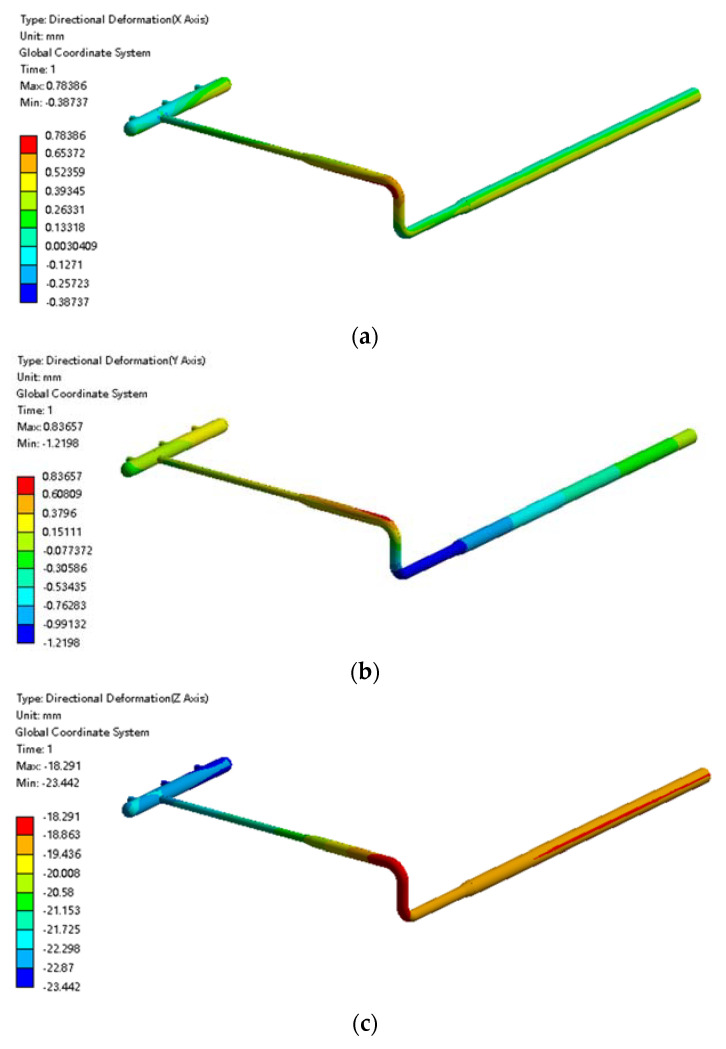
Cloud diagram of pipeline displacement at 0 °C: (**a**) cloud diagram of pipeline displacement at the x direction; (**b**) cloud diagram of pipeline displacement at the y direction; (**c**) cloud diagram of pipeline displacement at the z direction.

**Figure 23 materials-15-05795-f023:**
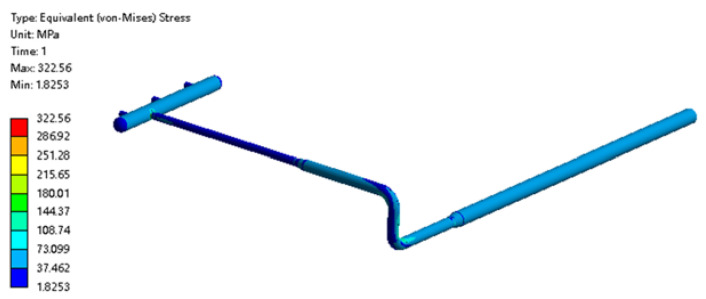
Cloud diagram of pipe stress distribution at −1 °C.

**Figure 24 materials-15-05795-f024:**
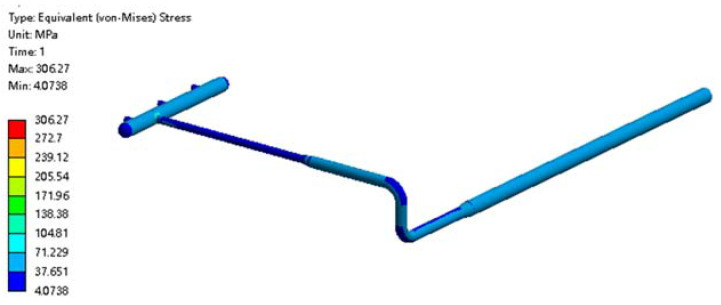
Cloud diagram of pipe stress distribution at 0 °C.

**Figure 25 materials-15-05795-f025:**
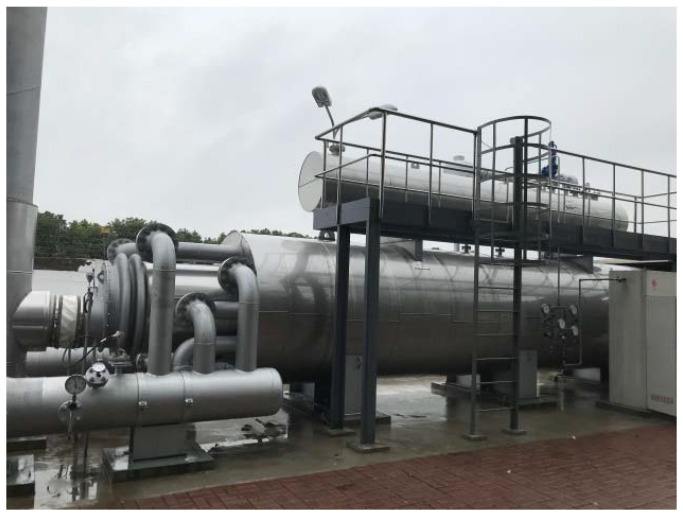
Main body diagram of the heating furnace.

**Table 1 materials-15-05795-t001:** Basic physical parameters of soils.

Soil Type	Density (kg/m^3^)	Elasticity Modulus (MPa)	Poisson’s Ratio	Angle of Internal Friction (°)	Cohesion (MPa)	Thermal Conductivity (W/m·°C)	Specific Heat Capacity (10^3^ kJ/m^3^·°C)
Unfrozen soil	1780	25	0.35	18.1	0.0516	1.36	1.326
Frozen soil	1700	45	0.25	15	1.32	1.89	1.516

**Table 2 materials-15-05795-t002:** Mechanical parameters of the steel pipeline.

Mechanical Parameters	Density (kg/m^3^)	Elasticity Modulus (MPa)	Poisson’s Ratio	Thermal Conductivity (W/m·°C)	Specific Heat Capacity (10^3^ kJ/m^3^·°C)	Linear Expansion Coefficient (1/K)
steel pipeline	7750	203,000	0.3	65.8	0.473	0.00001071

**Table 3 materials-15-05795-t003:** Mesh convergence results.

Number of Grid Nodes	Maximum Stress (MPa)	Deviation
1,005,453	1185.06	7.26%
1,121,153	1104.80	datum
1,285,438	1087.42	1.57%

## Data Availability

Not applicable.

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
