# Peer review of "Frost Heaving Damage Mechanism of a Buried Natural Gas Pipeline in a River and Creek Region"

_materials, 2022, doi:10.3390/ma15165795_

Round 1
Reviewer 1 Report
The authors studied the frost heaving damage mechanism of buried natural gas pipe line in river & creek region. The work can be published in journal. However major revision is necessary to improve the quality of the paper content.
1. The abstract should present the research findings.
2. The quality of images is poor. The authors should change Figures 2, 3, 4, 5, 6, 7, 8, 21-24.
3. The size of the scale bar on simulation figures should be increased. It is not clearly visible.
Author Response
Point 1: The abstract should present the research findings.
Response 1: Thanks very much for your comments, which are very helpful to improve the quality of this article. We have added some missing research results to the abstract.
Point 2: The quality of images is poor. The authors should change Figures 2, 3, 4, 5, 6, 7, 8, 21-24.
Response 2: We appreciate it very much for this good suggestion, and we have replaced the figures with a clear version.
Point 3: The size of the scale bar on simulation figures should be increased. It is not clearly visible.
Response 3: Thank you again for your positive comments and valuable suggestions to improve the quality of our manuscript.We have increased the size of the scale bar on simulation figures to make sure that they are clearly visible.
Reviewer 2 Report
All comments are in the uploaded pdf file

Author Response
Point 1: It should be necessary to include since the begining of the manuscript the pipeline material specification (e.g. API 5L X52, etc)
Response 1: Thanks very much for your comments, which are very helpful to improve the quality of this article. We have added the pipeline material specification in the first chapter.
Point 2: It is recommended to include some of the pipeline characteristics like: yield strength, UTS,steel chemical composition, grain size, etc
Response 2: We appreciate it very much for this good suggestion, and we have added some of the pipeline characteristics like: yield strength, UTS steel chemical composition, grain size, etc in chapter 3.1 Model verification.
Point 3: It is recommended to include other soil characteristics like soil texture and soil density because they can provide an idea of the soil type.
Response 2: Thank you again for your positive comments and valuable suggestions to improve the quality of our manuscript.We have added other soil characteristics in chapter 3.1 Model verification, in order to provide an idea of the soil type.
Point 4: after some time the steel pipeline tends to age. This is described by N.E. Gonzalez-Arevalo et al. in the paper "Influence of aging steel on pipeline burst pressure prediction and its impact on failure probability estimation". The steel grain size changes because of the aging. How this aging affect the thermal characteristic. How would it affect in your study for example
Response 4: Thanks very much for the comments, and we have carefully studied this paper. Natural gas pipelines are made of low carbon steel whose grain size varies with aging. With the increase of aging time, not only the pipeline will be corroded and deteriorated, but also the aging of the structure and the attenuation of mechanical properties are more serious. Although we agree that this is an important consideration, here we study the macroscopic deformation of gas pipelines without considering the microstructure changes.
Reviewer 3 Report
“Frost Heaving Damage Mechanism of Buried Natural Gas Pipe-line in River & Creek Region”
In this paper, the authors investigated the impact of frost swelling of soil on the natural gas high-pressure regulator station in the river & creek region. It is an interesting topic. In addition, the authors are suggested to address the following comments to meet the journal's requirements and be suitable for publication.
1) There are many typos in the manuscript that need to be fixed.
2) The novelty needs to be clear.
3) Very poor quality of figures, all of them need to be represented with at least 300 DPI.
4) I cannot see the legend of the figures.
5) A Mesh independence study must be involved in the numerical analysis.
6) Validation of your model with a similar previous study from literature needs to be added.
Author Response
Point 1: There are many typos in the manuscript that need to be fixed.
Response 1: We feel sorry for our carelessness. In our resubmitted manuscript, the typo is revised. Some typos may be caused by automatic line change during layout.Thanks for your correction.
Point 2: The novelty needs to be clear.
Response 2: We appreciate it very much for this good suggestion, and we have done it according to your ideas.
Point 3: Very poor quality of figures, all of them need to be represented with at least 300 DPI.
Response 3: Thanks very much for your comments. We are very sorry about the figures,and we have replaced the figures with a clear version.
Point 4: I cannot see the legend of the figures.
Response 4: We appreciate for your warm work earnestly, and we have confirmed with the editor that all the figures have the legend.
Point 5: A Mesh independence study must be involved in the numerical analysis.
Response 5: Thank you again for your positive comments and valuable suggestions to improve the quality of our manuscript, and we have added a mesh independence study in chapter 3.2 Study of mesh independence.
Point 6: Validation of your model with a similar previous study from literature needs to be added.
Response 6: Thanks very much for your comments, which are very helpful to improve the quality of this article. We have added the suggested content to the manuscript in chapter 3.1 Model validation.
Round 2
Reviewer 1 Report
The authors have incorporated all the corrections in the revised manuscript.
Author Response
Thanks a lot. We appreciate for editor and reviewer's warm work earnestly.
Reviewer 3 Report
“Frost Heaving Damage Mechanism of Buried Natural Gas Pipe-line in River & Creek Region”
In this paper, the authors investigated the impact of frost swelling of soil on the natural gas high-pressure regulator station in the river & creek region. It is an interesting topic. In addition, the authors are suggested to address the following comments to meet the journal's requirements and be suitable for publication.
1) Validation of your model with a similar previous study from literature needs to be added. I don’t find the validation in the revised paper (Figure or Table).
Hint: Any numerical study must validate your model with any similar or related study from literature.
